# Correlation Analysis between Intraocular Pressure and Extraocular Muscles Based on Orbital Magnetic Resonance T2 Mapping in Thyroid-Associated Ophthalmopathy Patients

**DOI:** 10.3390/jcm11143981

**Published:** 2022-07-08

**Authors:** Ban Luo, Wei Wang, Xinyu Li, Hong Zhang, Yaoli Zhang, Weikun Hu

**Affiliations:** 1Department of Ophthalmology, Tongji Hospital, Tongji Medical College, Huazhong University of Science and Technology, Wuhan 430074, China; banluoeye@hust.edu.cn (B.L.); wwnissan@163.com (W.W.); lixy07@126.com (X.L.); sgy804408@gmail.com (H.Z.); 2Department of Ophthalmology, Tianyou Hospital/Tongji Hospital, Wuhan University of Science and Technology, Wuhan 430072, China

**Keywords:** thyroid-associated ophthalmopathy, extraocular muscle, magnetic resonance imaging, intraocular pressure, T2 relaxation time

## Abstract

Background: The correlation between intraocular pressure (IOP) and the magnetic resonance imaging (MRI) parameters in thyroid-associated ophthalmopathy (TAO) patients was explored. Methods: This study included 82 eyes in 41 TAO patients who had a large difference in the IOP between each eye. We measured the T2 relaxation time (T2RT) of the extraocular muscles (EOMs), the orbital fat, and the area of the EOMs. Results: There was a positive correlation between IOP and exophthalmos, the clinical activity score (CAS), the T2RT (of the medial rectus (MR)), the area of the MR, inferior rectus (IR) and lateral rectus, and the mean area. We established a regression model with IOP as the dependent variable, and the area of the IR was statistically significant. Conclusions: High IOP in TAO patients was positively correlated with the degree of exophthalmos and EOM inflammation (especially the inferior rectus). The state of the EOMs in an orbital MRI may partially explain high IOP and provide the necessary clinical information for subsequent high IOP treatment.

## 1. Introduction

Thyroid-associated ophthalmopathy (TAO) is a chronic autoimmune inflammatory disease that affects the orbital fat (OF), extraocular muscles (EOMs), eyeball, and eye appendages, and is related to thyroid autoimmune pathology [1]. Clinical signs include widened palpebral fissure, eyelid retraction, conjunctival hyperemia, exophthalmos, corneal exposure, restrictive myopathy, optic neuropathy, and other symptoms [2]. In most patients, the effects on the eye appear mild, and the severe form of the disease affects 3% to 5% of individuals [3].

Many studies have shown that people with thyroid disease have a higher risk of high intraocular pressure (IOP) and glaucoma [4,5]. However, there have also been studies that were skeptical of these findings. A study by Cockerham showed that, while a quarter of TAO patients developed ocular hypertension, glaucoma damage was rare and even lower than the relative risk for glaucoma visual field defects in healthy people [5]. There are also differing opinions on the treatments for lowering the IOP in TAO patients with ocular hypertension. Despite the various theories so far, the relationship between ocular hypertension, open-angle glaucoma, and TAO is still a mystery. The increase in IOP in TAO may be caused by inflammation that causes swelling of the EOMs, which restricts and compresses the eyeball [6]; a decrease in orbital venous drainage leads to an increase in episcleral venous pressure [7], or an increase in mucopolysaccharide deposition in the trabecular meshwork, which thereby reduces the outflow of aqueous humor [8].

A study by Stoyanova et al. showed that a group of TAO patients with a larger thickness sum in the EOMs, as shown by orbital computed tomography, had higher IOP [9]. Studies have also found a negative correlation between IOP and the flow velocity of the supraocular vein in TAO patients [7]. Some studies have proposed that inflammation-induced EOM edema and inflammation-activated fibroblasts differentiate into adipocytes in TAO patients, leading to crowding of the orbital tissue and then increased IOP. However, there have been no studies to quantify this relationship.

By using the T2 relaxation time (T2RT), orbital magnetic resonance imaging (MRI) not only detects the presence or absence of swollen tissue, but also objectively and quantitatively evaluates the inflammatory activity of the orbital tissue in TAO patients [10,11]. The increase of T2 signal intensity in MRI may indicate edema changes which caused by autoimmune inflammation and/or vascular congestion, as the T2RT reflects the water content of the tissue. Orbital MRI can provide a more comprehensive assessment of TAO patients. [10].

Studies have found a variety of systemic and ocular risk factors associated with TAO combining with open-angle glaucoma, including old age, female gender, a family history of glaucoma, myopia, diabetes, hypertension, the presence of pseudoexfoliation and thyroxine treatment [12]. There are also many factors affecting the IOP of TAO patients. The current study included patients with high IOP in a unilateral eye and compared the differences in parameters, such as the EOMs and the clinical activity score (CAS) between both eyes; it studied the factors that lead to increased IOP in patients with TAO so as to provide information for the selection of subsequent treatment options. Furthermore, this study effectively avoided the influence of confounding factors, such as age, sex, blood pressure, measurement time, medication history, thyroid hormone status, and history of other systemic diseases among the individuals with IOP.

## 2. Materials and Methods

### 2.1. Subjects

A retrospective analysis was performed on clinical data from November 2015 to December 2019 for a total of 82 eyes in 41 TAO patients who met the inclusion criteria at Tongji Hospital, Tongji Medical College of Huazhong University of Science and Technology. The age, gender, thyroid disease status, vision, IOP, CAS, exophthalmos, and orbital MRI-T2 mapping scans of the patients were collected.

Inclusion criteria: TAO was diagnosed according to the EUGOGO guidelines [13]. An IOP difference ≥ 2 mmHg between the two eyes, an IOP ≤ 21 mmHg in a unilateral eye, and an IOP > 21 mmHg in the other eye, or an IOP difference ≥ 5 between the two eyes and an IOP ≤ 21 mmHg in the bilateral eyes were included. Exclusion criteria: patients with inconsistent local corticosteroid therapy in the eyes in the past 3 months, a history of orbital decompression and EOM surgery in the past year, or intervals between the IOP measurement and orbital MRI-T2 mapping examination of greater than 1 month were excluded.

According to the IOP of the bilateral eyes, the eyeball with the lower IOP was included in the low IOP group, and the other eye with the higher IOP was included in the high IOP group. The IOP was measured with a non-contact tonometer (NIDEK). TAO activity was evaluated according to the CAS. The contents of the CAS according to EUGOGO guidelines (2021) [13] include the following points, and each positive counts as 1 point for a total of 7 points: 1. Spontaneous retrobulbar pain; 2. Pain on attempted upward or downward gaze; 3. Redness of eyelids; 4. Redness of conjunctiva; 5. Swelling of caruncle or plica; 6. Swelling of eyelids; 7. Swelling of conjunctiva (chemosis).The CAS and the degree of exophthalmos were measured by the same orbital plastic surgeon (B.L.) All data were obtained by averaging them after three measurements. The data of the cross-sectional areas of the extraocular muscles and the T2 relaxation times of the orbital tissues were mainly measured by two radiologists who had worked for more than 5 years, and the average values of the data were used. The orbital MRI-T2 mapping reports were issued by the same imaging specialist (J.Zh.) for all patients.

### 2.2. Measurement of Orbital MRI-T2 Mapping Parameters

An orbital MRI was performed for all participants using a 3.0 T MRI system (Signa HDxt, GE Healthcare, Pittsburgh, PA, USA). The T2RTs of the EOMs were measured using a multi-slice multi-spin echo pulse sequence with a TR of 1500 ms, 7 TE values (22, 33, 44, 55, 66, 77, and 88 ms), a 180 × 180 mm field of view, 3.0 mm slice thickness, a 256 × 256 matrix, and 1 NEX. The color-coded T2 calculation was generated by a single exponential curve fitting using T2 mapping software (ADW4.4 workstation, GE Healthcare, Pittsburgh, PA, USA) (Figure 1a). The T2RTs (ms) and the areas (mm^2^) of five EOMs (inferior rectus (IR), superior rectus/levator complex (SRLCLC), medial rectus (MR), lateral rectus (LR), and superior oblique (SO)) were measured (Figure 1b). The maximum value for the T2RT or the area on the coronal section of each EOM was recorded as the final T2RT or area value.

### 2.3. Image Analysis

All the MR data from T2 mapping were processed using open-source software (Fire Voxel, New York University, New York, NY, USA) by two experienced neuroradiologists (approximately 4 and 7 years of clinical experience in head and neck radiology). The maximum cross-sectional area and the T2RT of the EOM were calculated for the five EOMs and orbital fat (superior, inferior, medial, lateral rectus, and superior oblique) within each orbit.

Blind to patient information, the two neuroradiologists selected the TE value that best delineated the EOM on the T2 mapping image, and carefully and independently drew a region of interest (ROI) layer by layer along the edge of the EOM, while avoiding orbital fat and air in the paranasal sinuses. Using partial volume effects to obtain the VOI (volume of interest) of the muscle, the same procedure was repeated to generate a VOI of five EOMs within each track. Due to the difficulty in separating the superior rectus and the levator superioris, they are delineated as a group of muscles called the superior rectus/levator complex. The orbital and extraocular muscles were delineated to delineate the boundary of the orbital fat, and the orbital fat VOI was obtained. The T2 relaxation time of each extraocular muscle was calculated using a single exponential T2 mapping fitting model. The value of T2 relaxation time was obtained by using S (TE) = S0 × e(−TE/T2), where S is the signal intensity (in arbitrary units), S0 is the initial signal intensity, TE is the echo time, and T2 is the T2 relaxation time (ms). The maximum cross-sectional area of each extraocular muscle was calculated using Image J (V1.8.0.112, National Institutes of Health, Bethesda, MD, USA).

### 2.4. Statistical Analysis

For statistical evaluation of the data, SPSS Statistics software (ver. 26, IBM, Armonk, NY, USA) was used for the statistical analysis. A Shapiro–Wilk test was used for the normality test. A paired *t*-test was used to compare the data that satisfied the normal distribution; if they did not, a Wilcoxon signed-rank test was used. The correlation analysis used a Spearman test. Taking the IOP as the dependent variable, a single-factor regression analysis combined with clinical experience was used to screen the independent variables, and the age, gender, CAS, EOM cross-sectional area, and T2RTs of the EOMs were included as independent variables for screening to build a variable linear regression model (*p* < 0.05). *p* < 0.05 was considered statistically significant. 

## 3. Results

The demographic characteristics of the TAO patients included in this study are shown in Table 1. 

The ocular data for the low IOP (LIOP) group and the high IOP (HIOP) group are shown in Table 2 and Figure 2. It can be seen that the values for the CAS, exophthalmos, T2RT of the MR, the area of the MR, IR and LR, the average area of the five EOMs and the total area of the five EOMs were significantly higher in the HIOP group than in the LIOP group (*p* = 0.001, *p* < 0.0001, *p* = 0.015, *p* = 0.002, *p* < 0.0001, *p* < 0.0001, *p* < 0.0001, *p* < 0.0001). In order to study the clinical characteristics of TAO eyes with ocular hypertension, we performed a statistical analysis on the clinical symptoms of TAO eyes. In terms of clinical symptoms, the incidence of restricted upward movement of the eye and spontaneous retrobulbar pain (y/n) were significantly increased in the HIOP group com-pared to the LIOP group (*p* = 0.0008, *p* = 0.0020).

### 3.1. Correlation Analysis

For the IOP, CAS, exophthalmos, T2RTs (ms), and the areas (mm^2^) of the five EOMs, the average T2 value and the average area of the EOMs were subjected to a bivariate correlation analysis, and it was found that IOP was positively correlated with the CAS, exophthalmos, T2RTs (MR), MR, IR, and LR areas, and the average area of the five EOMs (Table 3).

### 3.2. Multiple Linear Regression

Taking IOP as the dependent variable, the CAS, exophthalmos, T2RT (ms), the areas (mm^2^) of the five EOMs, and the T2RT of OF were used as independent variables for variable screening (*p* < 0.05), and finally, exophthalmos, the T2RT of MR, the area of IR and LR were included in the linear regression to build a regression model (Table 4).

Linear regression equation:IOP = 10.426 + 0.345 × exophthalmos − 0.057 × T2RT(MR)+ 0.103 × Area (IR) +0.050 × Area (LR)

The area of the IR muscle is statistically significant.

## 4. Discussion

In this study, the values for the CAS, exophthalmos, the T2RT of the MR, IR and LR areas, and the mean area in the higher IOP group were significantly higher than in the lower IOP group. The T2RTs and the areas of the EOMs were higher in the high IOP group, and there was a positive correlation between them. We suggest that there are several reasons for this. First, the thickening of the EOMs makes orbit tissue more crowded, which mechanically compresses the eyeball, resulting in increased IOP. Second, the increased volume increases the pressure of the scleral vein and affects the outflow of the aqueous humor, which also increases the IOP. A study by Seo et al. also showed that the postoperative IOP of TAO patients was significantly lower than their preoperative IOP [14], indicating that the increase in the volume of the orbital tissue did cause an increase in IOP. At the same time, some researchers have suspected that the deposition of hyaluronic acid in the trabecular meshwork of TAO patients could cause obstruction of the outflow of their aqueous humor. Studies have found that the trabecular meshwork was the target tissue of thyroid hormones and that T3 could regulate HA levels. This supports the view that thyroid hormones can regulate the levels of HA in the eyes and may affect the outflow of aqueous humor [8]. 

TAO is an autoimmune disease characterized by the inflammation of the orbital connective tissue. The orbital volume increases due to increased adipogenesis and the excessive production of glycosaminoglycans, as well as due to EOM fibrosis [15]. Our research found that the exophthalmos and CAS in the high IOP group were significantly higher than those in the low IOP group, and that the IOP and exophthalmos were positively correlated. We believe that high exophthalmos is one of the manifestations of hypertrophy in the intraorbital tissue because the increased tissue volume in the rigid wall of the orbit tends to push the eye forward, causing it to protrude, and the CAS can reflect the inflammation of the orbital tissue and the degree of orbital venous stasis caused by intraorbital crowding [16,17]. Therefore, more hypertrophic EOMs and greater proliferation of orbital adipose tissue will lead to greater exophthalmos and more likely cause higher IOP and CASs.

As mentioned in the introduction, the CAS is a subjective and qualitative measurement method, while an orbital MRI is an objective and quantitative measurement method. At the same time, the T2RT is more reliable in distinguishing the fibrosis expansion in the EOMs with lower disease activity from the inflammation and edema in the EOMs with higher disease activity. Our study quantitatively evaluated the EOM area and the orbital tissue T2RTs of TAO patients. We combined this with IOP measurements and used linear regression modeling to analyze the factors affecting IOP. This is more convincing than previous qualitative analyses.

In the correlation analysis, IOP had the strongest correlation with the IR area. When the regression model of the factors influencing IOP was established, the IR area became the only statistically significant EOM parameter. Previous studies have shown that the EOM most commonly affected by TAO is the IR [18]. We speculate that this may be because the most commonly involved muscle in TAO patients is the IR, which is more representative of the degree of orbital crowding than the other EOMs. At the same time, according to previous studies, the eye position while measuring IOP can also cause a temporary increase in the IOP measurement. For example, IOP is significantly higher when gazing upward than in the primary eye position, which is more pronounced in normal eyes than in TAO patients [19]. Studies on TAO IOP have shown that in a non-relaxed eye position, the infiltration and fibrosis of the EOMs can lead to an ultra-short-term increase in IOP caused by the pressure of EOMs on sclera [20]. At the same time, the study by Gomi et al. has proved that IOP can be significantly reduced by releasing the extraocular muscle restriction in TAO patients [6]. When we measured the IOP, the patients were in the primary position. Therefore, some patients with IR infiltration or fibrosis needed to resist the pulling of the IR, which will inevitably put pressure on the eyeball.

A shortcoming of our research is that a non-contact applanation tonometer was used, which could not measure the true IOP according to the patient’s primary eye position as well as handheld tonometers, such as iCare. At the same time, this study found that the measured value of the non-contact applanation tonometer was higher than that of the iCare contact tonometer and the Goldman tonometer [21]. Failure to measure the central corneal thickness is another shortcoming of this study. Because of the heterogeneity between the eyes, the influence of the central corneal thickness on the IOP results needs to be considered.

Some of our research results can provide useful information for clinical use. In all cases of high IOP, clinicians should look for other signs, as elevated IOP may simply be a manifestation of TAO. For high IOP without visual field and optic nerve fiber abnormalities, the cause should be judged carefully. For patients with transient IOP increases due to changes in gaze position, they can be temporarily treated, not for lowering their IOP, but for increased IOP from the crowding of the orbital contents caused by long-term inflammatory reactions; the visual field and the thickness of the nerve fiber layer can be monitored, and the IOP can be appropriately treated according to the situation. At the same time, studies have shown that when compared with glaucoma patients, the TAO 24-h IOP rhythm pattern is closer to that of healthy subjects [22]. Therefore, for those TAO patients with high IOP, a 24-h IOP measurement can also be considered for further judgment.

## 5. Conclusions

The high IOP in TAO patients is positively correlated with the degree of exophthalmos and EOM inflammation. The orbital MRI EOM state can explain part of the cause of high IOP and provide necessary clinical information for subsequent high IOP treatment.

## Figures and Tables

**Figure 1 jcm-11-03981-f001:**
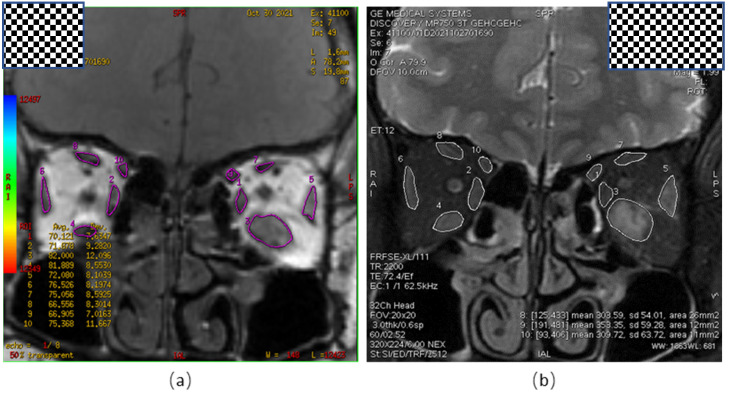
Measurement of Orbital MRI-T2 Mapping Parameters. (**a**) Orbital coronal pseudo-color map, representing the T2RT value of the ROI in each EOM; (**b**) orbital coronal view, indicating the largest cross-sectional area of the EOM. T2RT, T2 relaxation time; ROI, region of interest; EOM, extraocular muscles.

**Figure 2 jcm-11-03981-f002:**
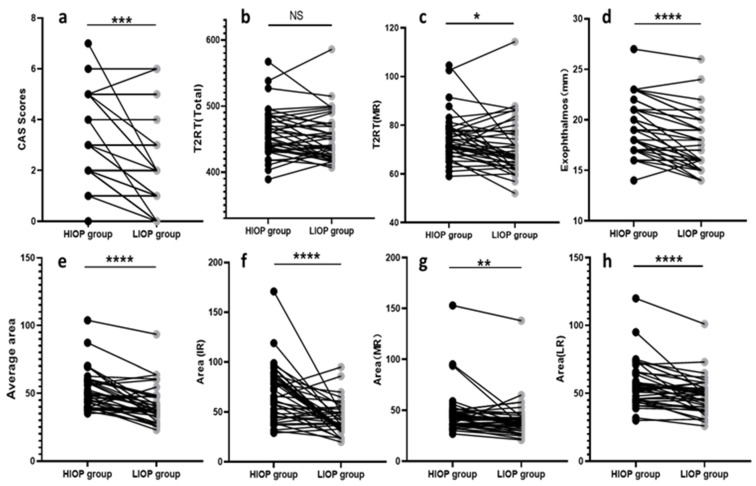
Wilcoxon signed-rank test charts show the differences of the CAS score (**a**), the overall orbital T2 relaxation time—T2RT (Total) (**b**), the medial rectus muscle T2 relaxation time—T2RT(MR) (**c**), exophthalmos (**d**), the average extraocular muscle cross-sectional area—average area (**e**), the inferior rectus cross-sectional area-area (IR) (**f**), the medial rectus cross-sectional area—area (MR) (**g**), and the lateral rectus cross-sectional area—area(LR) (**h**) between the HIOP and LIOP groups. CAS, clinical activity score; T2RT, T2 relaxation time; MR, medial rectus; IR, inferior rectus; LR, lateral rectus; HIOP, high intraocular pressure; LIOP, low intraocular pressure; NS, not statistically different; * *p* < 0.05; ** *p* < 0.01; *** *p* < 0.001; **** *p* < 0.0001.

**Table 1 jcm-11-03981-t001:** Demographic characteristics of TAO patients.

Demographic Characteristics	
Gender (Male/Female)	21/20
Age (Year)	44.59 ± 10.77
Thyroid function (euthyroid/hyperthyroidism/hypothyroidism)	8/32/1
History of glucocorticoid use (No/Yes)	33/8

**Table 2 jcm-11-03981-t002:** Ocular characteristics of TAO patients in 2 groups.

Parameters	LIOP Group (*n* = 41)	HIOP Group (*n* = 41)	*p* Value
CAS	1.90 ± 1.73	2.68 ± 1.77	0.001 ^&^
Exophthalmos	17.54 ± 2.76	19.40 ± 2.74	<0.0001 ^&^
T2RT (MR)	70.81 ± 11.25	74.68 ± 9.39	0.015 ^&^
Area (MR)	40.24 ± 18.40	46.78 ± 21.82	0.002 ^&^
T2RT (IR)	79.55 ± 7.67	79.26 ± 9.28	0.850 ^$^
Area (IR)	43.05 ± 16.28	68.78 ± 27.89	<0.0001 ^&^
T2RT (LR)	72.81 ± 6.52	74.38 ± 6.75	0.251 ^&^
Area (LR)	48.54 ± 13.34	56.93 ± 16.52	<0.0001 ^&^
T2RT (SRLC)	81.87 ± 15.38	82.96 ± 14.42	0.791 ^&^
Area (SRLC)	57.15 ± 32.20	65.17 ± 37.64	0.925 ^&^
T2RT (SO)	75.42 ± 6.84	75.92 ± 7.67	0.506 ^$^
Area (SO)	18.10 ± 5.03	20.08 ± 6.73	0.118 ^&^
T2RT (OF)	71.11 ± 14.09	70.51 ± 14.29	0.667 ^&^
Average T2RT	75.26 ± 6.02	76.29 ± 5.95	0.168 ^&^
Total T2RT	451.57 ± 36.13	457.71 ± 35.72	0.168 ^&^
Average area	41.47 ± 13.11	51.51 ± 14.01	<0.0001 ^&^
Total area	206.63 ± 66.42	257.54 ± 70.04	<0.0001 ^&^
RUME (y/n)	11/30	26/15	0.0008 ^#^
RDME (y/n)	9/32	10/31	0.794 ^#^
RIME (y/n)	0/41	1/40	>0.999 *
ROME (y/n)	6/35	9/32	0.392 ^#^
Spontaneous retrobulbar pain (y/n)	9/32	19/22	0.020 ^#^
Pain on attempted upward or downward gaze (y/n)	9/32	14/27	0.219 ^#^
Redness of eyelids (y/n)	9/32	10/31	0.794 ^#^
Redness of conjunctiva (y/n)	12/29	16/25	0.352 ^#^
Swelling of eyelids (y/n)	26/15	30/11	0.343 ^#^
Swelling of conjunctiva (y/n)	10/31	14/27	0.332 ^#^
Swelling of caruncle or plica (y/n)	3/38	7/34	0.177 ^#^

LIOP, low intraocular pressure; HIOP, high intraocular pressure;CAS, clinical activity score; T2RT, T2 relaxation time; MR, medial rectus; IR, inferior rectus; LR, lateral rectus; SRLC, superior rectus/levator complex; SO, superior oblique; OF, orbital fat; RUME, restricted upward movement of the eye; RDME, restricted downward movement of the eye; RIME, restricted inward movement of the eye; ROME, restricted outward movement of the eye; ^$^, paired *t*-test; ^&^, Wilcoxon signed-rank test; *, Fisher test; ^#^, chi-square test.

**Table 3 jcm-11-03981-t003:** Correlation analysis between IOP and CAS, exophthalmos and OMR parameters.

Parameters	Correlation Coefficient with IOP	*p* Value
CAS	0.254	0.021
Exophthalmos	0.402	<0.001
T2RT (MR)	0.250	0.023
Area (MR)	0.257	0.020
T2RT (IR)	−0.101	0.367
Area (IR)	0.550	<0.001
T2RT (LR)	0.093	0.406
Area (LR)	0.340	0.002
T2RT (SRLC)	0.034	0.759
Area (SRLC)	0.033	0.765
T2RT (SO)	0.045	0.691
Area (SO)	0.178	0.111
T2RT(OF)	0.106	0.344
Average T2RT	0.151	0.177
Average area	0.440	<0.001

IOP, intraocular pressure; OMR, ocular magnetic resonance; CAS, clinical activity score; T2RT, T2 relaxation time; MR, medial rectus; IR, inferior rectus; LR, lateral rectus; SRLC, superior rectus/levator complex; SO, superior oblique; OF, orbital fat.

**Table 4 jcm-11-03981-t004:** Model of liner regression.

	B	P
Constant	10.426	0.061
Exophthalmos	0.345	0.153
T2RT(MR)	−0.057	0.471
Area (IR)	0.103	0.001
Area (LR)	0.050	0.307

IOP = 10.426 + 0.345 × exophthalmos—0.057 × T2RT (MR)+ 0.103 × Area (IR) + 0.050 × Area (LR). T2RT, T2 relaxation time; MR, medial rectus; IR, inferior rectus; LR, lateral rectus.

## Data Availability

Not applicable.

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
