# Peer review of "Correlation Analysis between Intraocular Pressure and Extraocular Muscles Based on Orbital Magnetic Resonance T2 Mapping in Thyroid-Associated Ophthalmopathy Patients"

_jcm, 2022, doi:10.3390/jcm11143981_

Round 1

Reviewer 1 Report

The authors present an interesting study to an important subject. However, there are several issues which need to be addressed.

  • References are given with capital numbers, not in accordance with journal requirements.
  • Non-contact tonometry is less accurate, Goldmann Applanation tonometry would be preferred, in the best case in different gaze direction (straight, slight down gaze, up-gaze)
  • Is there any knowledge about the pachymetry of patients, since this can effect the IOP?
  • Was really only the superior rectus measured or the superior rectus/ levator complex?
  • Statistics: Was a correction performed for multiple testing, e.g. Bonferroni?
  • Results: Have you made a subgroup analysis of active vs inactive TAO? This would be really interesting to discern if CAS is directly involved in higher IOP by inflammatory mechanisms or just indirectly by more muscle swelling, exophthalmos etc.
  • IR was most prominent. In clinical practice we see a massive difference in IOP between slight down gaze and straight gaze in patients with IR swelling/ fibrosis. Therefore, it is unfortunate that the eye position during IOP measurement was not assessed.

Author Response

Dear reviewers.

Re: Manuscript ID: jcm-1731410 and Title:Correlation analysis between intraocular pressure and extraocular muscles based on orbital magnetic resonance T2 mapping in thyroid-associated ophthalmopathy patients

Thank you for your letter and for the reviewer’ comments concerning our manuscript entitled” Correlation analysis between intraocular pressure and extraocular muscles based on orbital magnetic resonance T2 mapping in thyroid-associated”(ID: jcm-1731410).Those comments are all valuable and very helpful for revising and improving our paper, as well as the important guiding significance to our researches. We have studied comments carefully and have made corrections in the paper and the responds to the reviewer’s comments are as following file.

Reviewer 2 Report

Zhang et al performed a retrospective study of TAO patients who had a large difference in IOP between each eye. The authors investigated the correlation of several metrics including IOP, T2 relaxation time of the extraocular muscles, orbital fat and the area of EOMs. The authors found a positive correlation between IOP and exophthalmos, clinical activity score, T2RT of the medial rectus, and the mean area. These changes in the EOMs may partially explain high IOP in TAO patients. Overall the experiments were well designed and the manuscript is reasonably written, with some minor typos. In order to improve the manuscript, the authors should further explain the following:

1.     Why would TAO patients experience unilateral high IOP? Is this typical?

2.     Further explain/define what “Clinical Activity Score” (CAS) is measuring/quantifying.

3.     Did either the low IOP eye or the high IOP eyes have any of the other clinical features of TAO such as widened palpebral fissure, eyelid retraction, conjunctival hyperemia, corneal exposure, restrictive myopathy, or other symptoms?

Author Response

Dear reviewer.

Re: Manuscript ID: jcm-1731410 and Title: Correlation analysis between intraocular pressure and extraocular muscles based on orbital magnetic resonance T2 mapping in thyroid-associated ophthalmopathy patients

Thank you for your letter and for the reviewer’ comments concerning our manuscript entitled” Correlation analysis between intraocular pressure and extraocular muscles based on orbital magnetic resonance T2 mapping in thyroid-associated”(ID: jcm-1731410).Those comments are all valuable and very helpful for revising and improving our paper, as well as the important guiding significance to our researchers. We have studied comments carefully and have made corrections in the paper and the responds to the reviewer’s comments are as following:

  1. Why would TAO patients experience unilateral high IOP? Is this typical?

Answer: TAO is the most common orbital disease. It mainly causes thickening of the extraocular muscles and the increase of orbital fat. The involvement of both eyes is often inconsistent, so the degree of disease in both eyes is inconsistent, and often manifests as proptosis, upper eyelid retraction, orbital inflammation, and limited eye movement. Inconsistencies, including inconsistencies in intraocular pressure. TAO with unilateral ocular hypertension is very common in TAO patients with ocular hypertension, but the specific incidence rate has not been reported. We collected 41 patients selected from 432 TAO patients who visited our hospital from 2015 to 2019. There are many factors that affect intraocular pressure, including age, gender, corneal thickness, and refractive status of the eye. In order to eliminate the influence of individual differences between patients on intraocular pressure, our study selected the eyes of TAO patients with unilateral intraocular hypertension as the research object.

  1. Further explain/define what “Clinical Activity Score” (CAS) is measuring/quantifying.

Answer: Thank you! The contents of the clinical activity score which refers to the TAO diagnosis and treatment guidelines formulated by EUGOGO in 2021 include the following points, each positive counts as 1 point, for a total of 7 points:1.Spontaneous retrobulbar pain2. Pain on attempted upward or downward gaze3. Redness of eyelids4. Redness of conjunctiva5. Swelling of caruncle or plica6. Swelling of eyelids 7. Swelling of conjunctiva (chemosis). we have been added these to the methods section of the article

  1. Did either the low IOP eye or the high IOP eyes have any of the other clinical features of TAO such as widened palpebral fissure, eyelid retraction, conjunctival hyperemia, corneal exposure, restrictive myopathy, or other symptoms?

Answer: Thank you for your helpful suggestion. We have done some statistical analysis on the clinical symptoms between TAO high intraocular pressure eye and low intraocular pressure eye, as shown in the following table. Relevant results have been added to Table 1 in the article. In terms of clinical symptoms, the  incidence of Restricted upward movement of the eye and Pain on attempt-ed upward or downward were significantly increased in HIOP group than in LIOP group. Maybe the high IOP eyes have the clinical features of TAO such as pain on attempt-ed upward or downward and restricted upward movement of the eye. These clinical symptoms can be caused by the limitation of the thickening inferior rectus muscle, so it is also confirmed that the high intraocular pressure of TAO is related to the thickening inferior rectus muscle. Our study did not collect information such as eye fissure widening, upper eyelid retraction, and corneal exposure, so no relevant statistical analysis was performed in this regard. These can be included in the future study.

parameters

LIOP group

(n=41)

HIOP group

(n=41)

P value

RUME (y/n)

26/15

11/30

0.0008

RDME (y/n)

10/31

9/32

0.794

RIME (y/n)

1/40

0/41

>0.999*

ROME (y/n)

9/32

6/35

0.392

Spontaneous retrobulbar pain(y/n)

14/27

9/32

0.219

Pain on attempted upward or downward gaze(y/n)

30/11

9/32

P<0.0001

Redness of eyelids(y/n)

10/31

9/32

0.794

Redness of conjunctiva(y/n)

16/25

12/29

0.352

Swelling of eyelids(y/n)

30/11

26/15

0.343

Swelling of conjunctiva(y/n)

14/27

10/31

0.332

Swelling of caruncle or plica(y/n)

7/34

3/38

0.177

RUME, restricted upward movement of the eye; RDME, restricted downward movement of the eye ;RIME, restricted inward movement of the eye; ROME, restricted outward movement of the eye *,Fisher test; Others are chi-square test.

Round 2

Reviewer 1 Report

The authors adressed my comments. Due to the retrospective nature of the study some points cannot be changed. However, the authors are undergoing a further prospective study to adress this issues. I am looking forward to the results in the future.